# Shift Invariance Can Reduce Adversarial Robustness

**Vasu Singla, Songwei Ge**[*]
Univeristy of Maryland
{vsingla, songweig}@cs.umd.edu

**Ronen Basri**
Weizmann Institute of Science
ronen.basri@weizmann.ac.il

**David Jacobs**
Univeristy of Maryland
dwj@cs.umd.edu

## Abstract

Shift invariance is a critical property of CNNs that improves performance on classification. However, we show that invariance to circular shifts can also lead to greater sensitivity to adversarial attacks. We first characterize the margin between classes when a shift-invariant *linear* classifier is used. We show that the margin can only depend on the DC component of the signals. Then, using results about infinitely wide networks, we show that in some simple cases, fully connected and shift-invariant neural networks produce linear decision boundaries. Using this, we prove that shift invariance in neural networks produces adversarial examples for the simple case of two classes, each consisting of a single image with a black or white dot on a gray background. This is more than a curiosity; we show empirically that with real datasets and realistic architectures, shift invariance reduces adversarial robustness. Finally, we describe initial experiments using synthetic data to probe the source of this connection.

## 1 Introduction

In *adversarial attacks* (Szegedy et al., 2013) against classifiers, an adversary with knowledge of the trained classifier makes small perturbations to a test image or even to objects in the world (Eykholt et al., 2018; Wu et al., 2020b) that change the output. Such attacks threaten the deployment of deep learning systems in many critical applications, from spam filtering to self-driving cars.

Despite a great deal of study, it remains unclear why neural networks are so susceptible to adversarial attacks. We show that invariance to circular shifts in Convolutional Neural Networks (CNNs) can be one cause of this lack of robustness. All reference to shifts will refer to circular shifts.

To motivate this conclusion we study in detail a simple example. Indeed, one of our contributions is to present perhaps the simplest possible example in which adversarial attacks can occur. Figure 1 shows a two class classification problem in which each class consists of a single image, a white or black dot on a gray background. We train either a fully connected (FC) network or a CNN that is designed to be fully shift-invariant to distinguish between them. Since each class contains only a single image, we measure adversarial robustness as the $l_2$ distance to an adversarial example produced by a DDN attack (Rony et al., 2019), using the training image as the starting point. The figure shows that the CNN is much less robust than the FC network, and that the robustness of the CNN drops precipitously with the image size.

In Sections 3 and 4 we explain this result theoretically. In Section 3 we study the effect of shift invariance on the margin of linear classifiers, for linearly separable data. We call a linear classifier

---

[*]Equal contribution

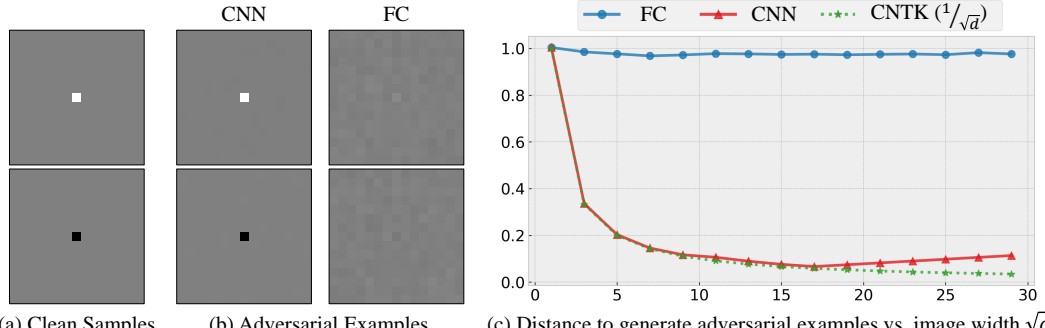

(a) Clean Samples     (b) Adversarial Examples     (c) Distance to generate adversarial examples vs. image width $\sqrt{d}$

Figure 1: In a binary classification problem, with each image as a distinct class (a), a FC network requires an average L2 distance close to 1 to construct an adversarial example while a shift-invariant CNN only requires approximately $\frac{1}{\sqrt{d}}$, where $d$ is the input dimension. (b)-left shows adversarial examples for the CNN. On the top, for example, the adversarial example looks just like the clean sample for the white dot class, but is classified as belonging to the black dot class. (b)-right shows that "adversarial" examples for the FC network just look like gray images. (c) shows accuracy under an adversarial attack, where the L2 norm of the allowed adversarial perturbation is shown in the horizontal axis. The curve labeled CNN shows results for a real trained network, while CNTK shows the theoretically derived prediction of $\frac{1}{\sqrt{d}}$ for an infinitely wide network.

shift invariant when it places all possible shifts of a signal in the same class. We prove that for a shift invariant linear classifier the margin will depend only on differences in the DC (constant) components of the training signals. It follows that for the two classes shown in Figure 1, the margin of a linear, shift invariant classifier will shrink in proportion to $\frac{1}{\sqrt{d}}$, where $d$ is the number of image pixels.

Next, in Section 4, we prove that under certain assumptions, a CNN whose architecture makes it shift invariant produces a linear decision boundary for the example in Figure 1, explaining its lack of adversarial robustness. We draw on recent work that characterizes infinitely wide neural networks as kernel methods, describing FC networks with the neural tangent kernel (NTK) (Jacot et al., 2018) and CNNs with a convolutional neural tangent kernel (CNTK) (Arora et al., 2019; Li et al., 2019). Our proof uses these kernels, and applies when the assumptions of this prior work hold. However, the results in Figure 1 are produced with real networks that do not satisfy these assumptions, suggesting that our results are more general.

We feel that it is valuable to produce such a simple example in which adversarial vulnerability provably occurs and can be fully understood. Still, it is reasonable to question whether shift invariance can affect robustness in networks trained on real data. In Section 5 we show experimentally that it can.

First, we compare the robustness of FC and shift-invariant networks on MNIST and Fashion-MNIST. On these smaller datasets, FC networks attain reasonably good test accuracy on clean data. We find that indeed, FC networks are also quite a bit more robust than shift-invariant CNNs. We also show that if we reduce the degree of shift invariance of a CNN, we increase its adversarial robustness on these datasets. Next we show that on the SVHN dataset, FC networks are more robust than ResNet architectures suggesting that this phenomenon extends to realistic convolutional networks. Finally, we consider industrial strength networks trained on CIFAR-10 and ImageNet. We note that ResNet and VGG are shift-invariant to a degree, while AlexNet, FCNs and transformer based networks are much less so. This suggests that ResNet and VGG will also be less robust than AlexNet, FCNs, and transformer networks, which we confirm experimentally.

Finally, we use simple synthetic data to explore the connection between shift invariance and robustness. We show that when shift invariance effectively increases the dimensionality of the training data, it produces less robustness, and that this does not occur when shift invariance does not increase dimensionality. This is in line with recent studies that connect higher dimensionality to reduced robustness. However, we also show, at least in a simple example, that higher dimensionality only affects robustness when it also affects the margin of the data.

**The main contribution of our paper is to show theoretically that shift invariance can undermine adversarial robustness, and to show experimentally that this is indeed the case on real-world networks.**. This enhances our understanding of robustness, and suggests that we can improve

robustness by judiciously reducing invariances that are not needed for good performance. Our code can be found at `https://github.com/SongweiGe/shift-invariance-adv-robustness`.

## 2 Related Work

In this section we survey prior work on the origin of adversarial examples and briefly discuss work on shift-invariance in CNNs, and on the NTKs.

### 2.1 The Cause of Adversarial Examples

One common explanation for adversarial examples lies in the curse of dimensionality. Goodfellow et al. (2015) argues that even with a linear classifier a lack of adversarial robustness can be explained by high input dimension. They suggest that large gradients naturally occur as the input dimension increases, assuming that the magnitude of weights remains constant. Following this, (Simon-Gabriel et al., 2019) provides theoretical and empirical evidence that gradient magnitude in neural networks grows as $\mathcal{O}(\sqrt{d})$ instead of $\mathcal{O}(d)$, where $d$ is the input dimension. Gilmer et al. (2018) analyzes the case of two classes consisting of concentric spheres, and shows that adversarial examples arise when the spheres are of high dimension. Shafahi et al. (2019) provides a general discussion of how high dimensional data can lead to adversarial examples can be inevitable by showing that a small perturbation in a high dimensional space can cover the whole space. Khoury & Hadfield-Menell (2019) argues theoretically and empirically that adversarial examples arise when data lies on a lower dimensional manifold in a high dimensional space.

Other studies show that adversarial examples could arise due to properties of the data. Wang et al. (2020) argues that adversarial examples can be explained by the dependence of the model on the high frequency components of the image data. Tsipras et al. (2019) shows that to achieve high accuracy one must sacrifice robustness due to biases in the features. Follow-up work (Schmidt et al., 2018) shows that the sample complexity requirements for an adversarially robust classifier to achieve good generalization are much higher than for a normal classifier. Furthermore, a recent study (Chen et al., 2020) shows that in some cases more data actually undermines adversarial robustness and also provides a more complete picture of this line of research.

Apart from the data, some papers look for explanations in the model. Nakkiran (2019) conjectures that the reason adversarial robustness is hard to achieve is that current classifiers are still not complex enough. Shah et al. (2020) shows that in many cases networks learn simple decision boundaries, with small margins, rather than more complex ones with large margins, which induces adversarial vulnerability. Daniely & Shacham (2020) shows that for most ReLU networks with random weights, most samples $x$ admit an adversarial perturbation at distance $O(\frac{\|x\|}{d})$, where $d$ is the input dimension. Shamir et al. (2019) shows that for ReLU networks, adversarial examples arise as a consequence of the geometry of $\mathbb{R}^n$ and show they can find targeted adversarial examples with $m\ l_0$ distance, when networks are designed to distinguish between $m$ classes. Galloway et al. (2019) provides empirical evidence that using batch normalization can reduce adversarial robustness. Madry et al. (2018) and Wu et al. (2020a) have discussed the relationship between robustness and network width. Kamath et al. (2020) examine the interplay between rotation invariance and robustness. They empirically show that by inducing more rotational invariance using data augmentation in CNNs and Group-equivariant CNNs the adversarial robustness of these models decreases. This work is especially relevant to our results that shift invariance can reduce robustness (A quite different notion of invariance is also explored in Jacobsen et al. (2019) and subsequent work).

### 2.2 CNNs and Shift-Invariance

Several papers have pointed out that the modern CNN architectures are not fully shift invariant for reasons such as padding mode, striding, and padding size. (Zhang, 2019) improves the local shift invariance to global invariance with the idea of "antialiasing". (Kayhan & Gemert, 2020) proposes to undermine the ability of CNNs to exploit the absolute location bias by using full convolution. (Chaman & Dokmanic, 2021) proposes the use of a novel sub-sampling scheme to remove the effects of down-sampling caused by strides and achieve perfect shift invariance. (Alsallakh et al., 2021) show padding mechanism can cause spatial bias in CNNs. (Azulay & Weiss, 2019) relates a lack of shift invariance to a failure to respect the sampling theorem.

## 2.3 Neural Tangent Kernel

Neural networks are usually highly non-convex. But recent studies have shown that when the networks are over-parameterized, their parameters stay close to the initialization during the training, which makes the analysis of the convergence and generalization of neural networks tractable (Allen-Zhu et al., 2019a,b; Du et al., 2018, 2019a). When optimizing the quadratic loss using gradient descent, the dynamics of an infinitely wide neural network can be well described by a kernel regression, using the *Neural Tangent Kernel* (NTK) (Jacot et al., 2018). This kernel only depends on the network architecture and initialization. Subsequent studies extend NTK beyond the fully connected network to various well-known architectures (Yang, 2020) including using the CNTK kernel for Convolutional Neural Networks (Arora et al., 2019; Li et al., 2019), the GNTK for Graph Neural Networks (Du et al., 2019b), and the RNTK for Recurrent Neural Networks (Alemohammad et al., 2021). Earlier works with these kernels have shown a superior performance over traditional kernels (Arora et al., 2020) while later theoretical and empirical studies show that NTKs are closely related to traditional kernels (Geifman et al., 2020; Chen & Xu, 2021).

The salient advantage of NTK is its ability as a tool to deliver tractable analysis of deep neural networks. Many interesting studies have been done to demystify the puzzling behaviors of deep neural networks using this tool. (Basri et al., 2019; Rahaman et al., 2019; Basri et al., 2020) study the inductive bias of the neural network in terms of the frequencies of the learned function. (Tancik et al., 2020) further apply these discoveries and random fourier features to empower neural networks to learn high-frequency signals more easily. (Huang et al., 2020) uses NTK to understand the importance of the residual connection proposed in ResNet (He et al., 2016). In this paper, we use NTK and CNTK to help us understand the behavior of FC and Convolutional Neural Networks in toy examples and draw a connection between CNN and adversarial examples.

## 3  Shift invariance and the linear margin

In this section we consider what happens when a linear classifier is required to be shift invariant. That is, we suppose that any circularly shifted version of a training example should be labeled the same as the original sample. For convenience of notation we consider classes of 1D signals of length $d$, in which $\mathbf{x} = (x_1, x_2, ...x_d)$. Our results are easily extended to signals of any dimension. We denote the set of training samples with label $+1$ by $X_1$, and the set of samples with label $-1$ by $X_2$. Let $\mathbf{x}^s$ denote the signal $\mathbf{x}$ shifted by $s$, with $0 \leq s \leq d - 1$. That is: $\mathbf{x}^s = (x_{s+1}, x_{s+2}, ...x_d, x_1, ...x_s)$. For a signal $\mathbf{x}$, let $\mathcal{SH}(\mathbf{x})$ denote the set containing every shifted version of $\mathbf{x}$. That is $\mathbf{x}^s \in \mathcal{SH}(\mathbf{x})$, $\forall s, 0 \leq s \leq d - 1$. Let $S_i = \cup_{\mathbf{x} \in X_i} \mathcal{SH}(\mathbf{x})$ denote the set of all shifted versions of all signals in the set $X_i$. Further, let $f_{dc}(\mathbf{x})$ denote the DC component of a signal. That is,

$$f_{dc}(\mathbf{x}) = \frac{1}{\sqrt{d}} \sum_{i=1}^{d} x_i$$

We denote $\bar{\mathbf{w}} = \frac{1}{\sqrt{d}} \mathbf{1}_d$. where $\mathbf{1}_d$ is a $d$-dimensional vector of all ones, so $f_{dc}(\mathbf{x}) = \bar{\mathbf{w}}^T \mathbf{x}$. We prove the following theorem:

**Theorem 1.** *Let $S_1$ and $S_2$ denote the sets of all shifts of $X_1$ and $X_2$, as described above. They are linearly separable if and only if $\max_{\mathbf{x_1} \in S_1} f_{dc}(\mathbf{x_1}) < \min_{\mathbf{x_2} \in S_2} f_{dc}(\mathbf{x_2})$ or $\max_{\mathbf{x_2} \in S_2} f_{dc}(\mathbf{x_2}) < \min_{\mathbf{x_1} \in S_1} f_{dc}(\mathbf{x_1})$. Furthermore, if the two classes are linearly separable then, if the first inequality holds, the margin is $\min_{\mathbf{x_2} \in S_2} f_{dc}(\mathbf{x_2}) - \max_{\mathbf{x_1} \in S_1} f_{dc}(\mathbf{x_1})$, and similarly if the second inequality holds. Furthermore, the max margin separating hyperplane has a normal of $\bar{\mathbf{w}}$.*

This theorem shows that a shift invariant linear classifier can only use the DC components of signals to separate them. The proof of this theorem is in the supplemental material. Here we will give an intuitive explanation for this result. For simplicity, we do not consider the bias term here.

When the two classes are linearly separable, this means WLOG that there exists a unit vector $\mathbf{w}$ such that $\mathbf{w} \cdot \mathbf{x_1^{s_1}} < \mathbf{w} \cdot \mathbf{x_2^{s_2}}, \forall \mathbf{x_1} \in X_1, \mathbf{x_2} \in X_2$ and any shifts $s_1$ and $s_2$. Consider the margin that arises when we consider just the sets $\mathcal{SH}(\mathbf{x_1})$ and $\mathcal{SH}(\mathbf{x_2})$. First, consider what happens when we compute $\mathbf{w} \cdot \mathbf{x_1}^s$ for all shifts, $s$. When $\mathbf{w} = \bar{\mathbf{w}}$, this inner product is constant and equal to $f_{dc}(\mathbf{x_1})$. Because $\bar{\mathbf{w}}$ is a constant vector, shifting $\mathbf{x_1}$ has no effect on the inner product with $\bar{\mathbf{w}}$. Similarly, $\bar{\mathbf{w}} \cdot \mathbf{x_2}^s = f_{dc}(\mathbf{x_2})$ for all shifts. So the margin between $\mathcal{SH}(\mathbf{x_1})$ and $\mathcal{SH}(\mathbf{x_2})$ will be $f_{dc}(\mathbf{x_2}) - f_{dc}(\mathbf{x_1})$.

For any other choice of $\mathbf{w}$, $\mathbf{w} \cdot \mathbf{x_1}^s$ will not generally be constant as $\mathbf{x_1}$ shifts. We can show that the average value of $\mathbf{w} \cdot \mathbf{x_1}^s$ over all shifts, $s$, will be $f_{dc}(\mathbf{x_1})f_{dc}(\mathbf{w})$. In this case, the margin between $\mathcal{SH}(\mathbf{x_1})$ and $\mathcal{SH}(\mathbf{x_2})$ must be no larger than the difference between the average values of $\mathbf{w} \cdot \mathbf{x_1}^s$ and $\mathbf{w} \cdot \mathbf{x_2}^s$, which is $f_{dc}(\mathbf{x_2})f_{dc}(\mathbf{w}) - f_{dc}(\mathbf{x_1})f_{dc}(\mathbf{w})$. Note that since $\mathbf{w}$ is a unit vector, $f_{dc}(\mathbf{w}) \leq 1$, and equals 1 for $\mathbf{w} = \bar{\mathbf{w}}$. Therefore, $\bar{\mathbf{w}}$ maximizes the margin for two reasons. First, the margin is scaled by $f_{dc}(\mathbf{w})$, which is maximized for $\mathbf{w} = \bar{\mathbf{w}}$. Second the values of $\mathbf{w} \cdot \mathbf{x_1}^s$ are constant for all choices of $s$ when $\mathbf{w} = \bar{\mathbf{w}}$, but can vary for other choices of $\mathbf{w}$. This variation can only reduce the margin.

We now consider the example shown in Figure 1. Let $\mathbf{x_1}$ denote an image consisting of a single 1 in a background of 0s, and let $\mathbf{x_2}$ denote an image containing a single -1 with 0s. Since $f_{dc}(\mathbf{x_1}) = \frac{1}{\sqrt{d}}$ and $f_{dc}(\mathbf{x_2}) = -\frac{1}{\sqrt{d}}$, if we let $X_1 = \{\mathbf{x_1}\}$ and $X_2 = \{\mathbf{x_2}\}$ it is straightforward to show:

**Corollary 1.** *$X_1$ and $X_2$ defined above are linearly separable with a max margin of 2. $\mathcal{SH}(X_1)$ and $\mathcal{SH}(X_2)$ are linearly separable with a max margin of $\frac{2}{\sqrt{d}}$ and a separating hyperplane with the normal vector $\bar{\mathbf{w}}$.*

# 4 Shift invariance and adversarial attacks: NTK vs. CNTK

To understand the robustness of shift invariant neural networks we examine the behavior of the neural tangent kernel (NTK) for two networks with simple inputs. It has been shown that neural networks with infinite width (or convolutional networks with infinite number of channels) behave like kernel regression with the family of kernels called NTKs (Jacot et al., 2018). Here we consider the NTK for a two-layer, bias-free fully connected network, denoted FC-NTK, and for a two-layer bias-free convolutional network with global average pooling, denoted CNTK-GAP (Li et al., 2019).

In our theorem below we use both FC-NTK and CNTK-GAP to construct two-class classifiers with a training set composed of two antipodal training points, $\mathbf{x}, -\mathbf{x} \in \mathbb{R}^d$. In both cases the kernels produce linear separators, but while FC-NTK produces a separator with constant margin, independent of input dimension, due to shift invariance CNTK-GAP results in a margin that decreases with $2/\sqrt{d}$, consistent with our results in Figure 1. Due to space considerations, the definition of NTK, CNTK and their corresponding network architectures, the minimum norm interpolant $g_k(\mathbf{z})$ of the kernel regression with kernel $k$, along with the proof of the theorem below are deferred to Section 8 in the Appendix.

**Theorem 2.** *Let $\mathbf{x}, -\mathbf{x} \in \mathbb{R}^d$ be two training vectors with class labels $1, -1$ respectively.*

1. *Let $k(\mathbf{z}, \mathbf{x})$ denote NTK for the bias-free, two-layer fully connected network. Then $\forall \mathbf{z} \in \mathbb{R}^d$, the minimum norm interpolant $g_k(\mathbf{z}) \geq 0$ iff $\mathbf{z}^T\mathbf{x} \geq 0$.*

2. *Let $K(\mathbf{z}, \mathbf{x})$ denote CNTK-GAP for the bias-free, two-layer convolutional network , and assume $H_K$ is invertible. Then $\forall \mathbf{z} \in \mathbb{R}^d$, either $g_K(\mathbf{z}) \geq 0$ iff $\mathbf{z}^T\mathbf{1}_d \geq 0$ or $g_K(\mathbf{z}) \geq 0$ iff $\mathbf{z}^T\mathbf{1}_d \leq 0$. (I.e., $\mathbf{z}^T\mathbf{1}_d = 0$ forms a separating hyperplane.)*

The theorem tells us that NTK and CNTK produce linear classifiers. (1) tells us that NTK produces a separating hyperplane with a normal vector $\mathbf{x}$, while (2) says that for CNTK the normal direction is $\mathbf{1}_d$. The following corollary follows directly from Thm 1, which explains the results in Figure 1.

**Corollary 2.** *With a training set composed of an antipodal pair the margin obtained with CNTK is the difference between their DC components.*

This theorem is proven in the supplemental material. Here we provide some intuition to explain the results. For the first part of the theorem, recall that NTK is a kernel method. When we use only two training examples, $\mathbf{x}$ and $-\mathbf{x}$, with labels 1 and -1, we will learn a function that sums two instances of the kernel centered at $\mathbf{x}$ and $-\mathbf{x}$. By symmetry, we expect the weight of one to be the negative of the weight of the other. Note that this holds only if $k(\mathbf{x}, \mathbf{x}) = k(-\mathbf{x}, -\mathbf{x})$, which is not true for all kernels, but is true for NTK. The kernel prediction then depends on the difference function, $k(\mathbf{x}, \mathbf{z}) - k(-\mathbf{x}, \mathbf{z})$. We show that for NTK this difference function is linear in $\mathbf{z}$, which implies that the decision boundary is linear. By symmetry, this linear decision boundary has a normal vector in the direction of $\mathbf{x}$. So any $\mathbf{z}$ will belong to the same class as $\mathbf{x}$ when $\mathbf{z}^T\mathbf{x} \geq 0$.

For the second part of the theorem, the two-layer CNTK takes the average of $k(\bar{\mathbf{z}}_i, \bar{\mathbf{x}}_j)$ over all patches. Here $k$ denotes the NTK and $\bar{\mathbf{z}}_i, \bar{\mathbf{x}}_j$ indicate patches of a fixed size from the input. Therefore

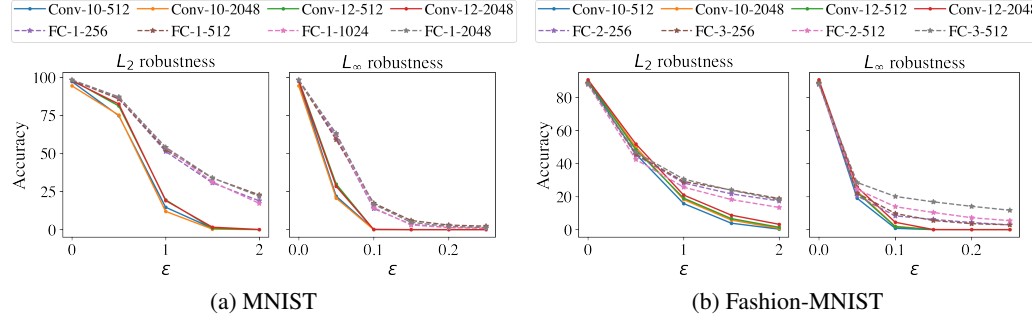

Figure 2: Robustness of Shift Invariant CNNs vs FC networks

| FC-1-256 | FC-1-512 | FC-1-1024 | FC-1-2048 | FC-2-256 | FC-3-256 | FC-2-512 | FC-3-512 |
|----------|----------|-----------|-----------|----------|----------|----------|----------|
| 16.00 | 20.69 | 20.69 | 19.20 | 19.46 | 18.19 | 20.68 | 18.38 |

Table 1: Shift Consistency of FCNs.

the difference $K(\mathbf{z}, \mathbf{x}) - K(\mathbf{z}, -\mathbf{x})$ is linear (an average of linear functions in $\mathbf{z}$), and due to shift-invariance, it must be orthogonal to the constant vector $\bar{\mathbf{w}}$.

## 5 Experiments

We have shown theoretically that shift invariance can reduce adversarial robustness. In this section we describe experiments that indicate that this does occur with real datasets and network architectures. We also provide experiments on simple, synthetic datasets to begin to address the question of why this might occur.

### 5.1 Real architectures and data

We first compare fully shift invariant and FC networks on the small datasets, MNIST and Fashion-MNIST. We then compare more realistic ResNet architectures with FC networks on the SVHN dataset. Finally we examine additional real architectures on CIFAR-10 and ImageNet.

#### 5.1.1 Shift-invariant CNNs vs. FC Networks

In this section we compare the adversarial robustness of a *fully shift-invariant convolutional neural network (CNN) with a fully connected network (FC)*.

**Datasets** We consider two datasets in which FC networks are able to attain reasonable performance compared to CNNs, MNIST (LeCun et al., 2010) and Fashion-MNIST (Xiao et al., 2017).

**Experimental Settings** All models were trained for 20 epochs using the ADAM optimizer, with a batch size of 200 and learning rate of 0.01. The learning rate is decreased by a factor of 10 at the 10th and 15th epoch. To evaluate the robustness of these models, we use PGD $l_2$ and $l_\infty$ attacks (Madry et al., 2018) with different $\epsilon$ values, a single random restart, 10 iterations and step-size of $\epsilon/5$.

We use various FC and shift invariant CNN networks for our experiments. The notation FC-X-Y denotes an FC network with X hidden layers and Y units in each hidden layer. Conv-X-Y denotes a network with a single convolutional layer with X kernel size and Y filters with stride 1 and circular padding. All networks use ReLU activation. CNNs have a penultimate layer that performs Global Average Pooling. Finally, a fully connected layer is applied at the end of all the CNNs, which outputs the final logits. **These CNN networks are born fully shift invariant.** On the contrary, the FC networks are inherently not shift invariant. We show this by calculating the consistency scores (Zhang, 2019) in Table 1 - the consistency calculates the percentage of the time that the model preserves its predicted labels when a random shift is applied to the image. We note that these scores are much smaller than 100 for CNNs. We report results for clean[2] and robust test accuracy for $l_2$ and $l_\infty$ attacks at different $\epsilon$ values in Figure 2a and 2b for MNIST and Fashion-MNIST respectively.

---

[2]Clean Accuracy of models is shown by $\epsilon = 0$.

**We observe that although the clean accuracy for the models is similar on the datasets, FC networks are more robust than Shift Invariant CNNs, especially for large $\epsilon$ values.**

### 5.1.2 CNNs with different levels of shift invariance

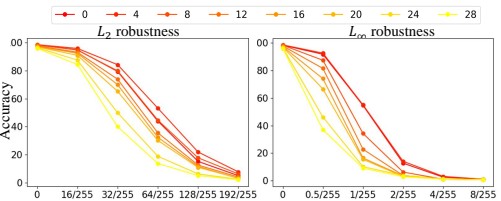

| padding size | 0 | 4 | 8 | 12 |
|---|---|---|---|---|
| consistency | 18.06 | 18.08 | 15.13 | 21.67 |
| padding size | 16 | 20 | 24 | 28 |
| consistency | 29.83 | 38.40 | 71.45 | 97.80 |

Table 2: Consistency to the shift of CNNs with different padding sizes.

Figure 3: Robustness of CNNs with different levels of shift invariance.

In this section we compare the adversarial robustness of convolutional neural networks (CNNs) that have different levels of shift invariance. Specifically, we train identical CNNs that all have a single convolutional layer with 1000 filters of kernel size 28 on the MNIST dataset and control the shift invariance through the size of circular paddings. Note that when the padding size equals the kernel size-1, i.e. in the "same" mode (Kayhan & Gemert, 2020), the output of convolution preserves the width and height of the input and the CNN is fully shift-invariant, which we will discuss more in the next section. When no padding is added, the model is just a FC network with 1000 hidden units. The higher the padding, the more shift invariant the model is. We demonstrate this by measuring the consistency of network output to shifts as suggested by (Zhang, 2019). As shown in Table 2, the large padding size does indicate a larger consistency to shifts. Next, we evaluate the robustness of these models using Method PGD $l_2$ and $l_\infty$ attacks (Madry et al., 2018) with different $\epsilon$ values. The results are shown in Figure 3, which illustrates that **the more shift invariant CNNs (yellowish lines) are less robust than the less shift invariant CNNs (reddish lines).**

### 5.1.3 Realistic architectures and shift invariance

Before describing experiments with real-world architectures, we discuss their shift invariance. With certain assumptions on the padding, (Cohen & Welling, 2016) proves that the convolution operation is equivariant to shifts. The convolution operation together with global pooling leads to shift invariance. Also, it is easy to derive from (Cohen & Welling, 2016) results for models that use stride and local pooling, which we summarize in the note below.

**Note 1.** *A Convolutional Neural Network that meets the following assumptions has some shift invariance:*

1. *It consists of $N$ fully convolutional or local pooling layers and $1$ global pooling layer followed by any fully connected layers.*

2. *Circular padding is used in a "same" mode in the convolutional and pooling layers.*

*Such a CNN with the strides $p_1, p_2, \cdots, p_N$ in the convolutional or pooling layers are invariant to a shift of $P$ pixels, where $P$ is any integer multiple of $\prod_{i=1}^{N} p_i$.*

(Kayhan & Gemert, 2020) discusses padding modes. Architectures applied to real problems are generally not born shift invariant (Zhang, 2019; Kayhan & Gemert, 2020) due to the violations of these assumptions. The most common case is the use of zero padding. Zhang (2019) has shown that realistic architectures with zero padding still preserve approximate invariance to shifts after training. Another common cause of a more severe lack of shift invariance is the use of a fully connected layer as a substitute for the global pooling layer. This is widely seen in the earlier network architectures such as LeNet (LeCun et al., 1989) and AlexNet (Krizhevsky, 2014) while more recent ones have universally adopted a global pooling layer (e.g. ResNet (He et al., 2016), DenseNet (Huang et al., 2017)) to reduce the number of parameters. Specifically, (Zhang, 2019) shows that in practice AlexNet is much less shift invariant than the other architectures. For this reason, we pay extra attention to the comparison of robustness between AlexNet and other architectures in the following sections when FC networks do not achieve decent state of the art accuracy.

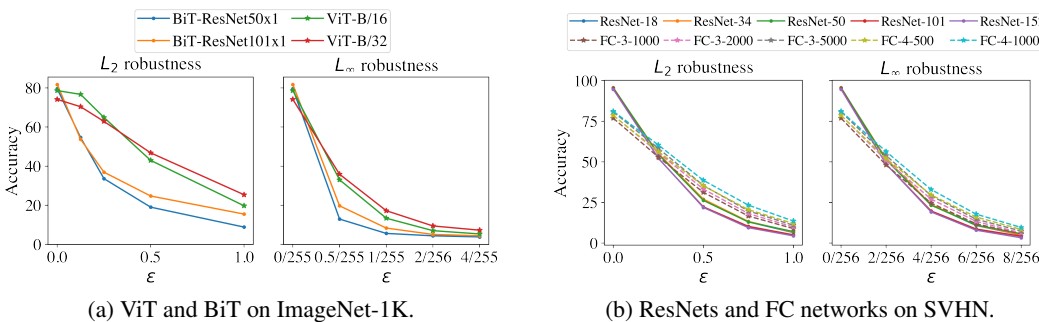

(a) ViT and BiT on ImageNet-1K.

(b) ResNets and FC networks on SVHN.

Figure 4: Robustness of Transformers, Resnets and FC networks.

### 5.1.4 ResNets vs. FC Nets

We compare ResNets with previously introduced FC networks on SVHN (Netzer et al., 2011). ResNets are real-world architectures that are not completely shift invariant. We use the SVHN dataset, since it is more difficult to attain reasonable performance with shallow networks on this dataset. The models were trained for 100 epochs using SGD with 0.9 momentum and batch-size of 128. A learning rate of 0.1, with decay of factor 10 at the 50th and 75th epochs was used. We use the same PGD attacker as described above. The results for clean and robust accuracy with different $\epsilon$ values for $l_2$ and $l_\infty$ are given in Figure 4b. **The ResNets have much higher clean accuracy, but lower accuracy on adversarial examples, especially when $\epsilon$ is large.** We conclude that ResNet's approximate shift invariance is associated with less robust performance.

### 5.1.5 More realistic networks

In this section we look at the robustness of trained networks on large scale datasets. We do not have a firm theoretical basis for comparing the shift-invariance of these networks, since none are truly (non-) shift invariant. But we do expect that AlexNet is less shift-invariant, due to violation of most of the assumptions in Note 1, and so we expect it to be more robust. Zhang (2019) introduces an empirical consistency measure of relative shift invariance, i.e. the percentage of the time that the model classifies two different shifts of the same image to be the same class, which we use to distinguish among models. Although it is not clear that this fully captures our notion of shift invariance, it does support the view that AlexNet is by far the least shift-invariant large-scale CNN. Specifically, the consistency of all the models except for AlexNet are over 85, while the consistency of AlexNet is only 78.11 (Zhang, 2019). We apply $l_2$ and $l_\infty$ PGD attacks to the pretrained models provided in the Pytorch model zoo on ImageNet. The accuracy of a few representative models under different attack strengths are shown in Figure 5b. **It clearly illustrates that AlexNet is indeed an outlier in terms of both shift invariance and robustness**. We see similar results with other models. (Galloway et al., 2019) suggests that batch normalization is a cause of adversarial examples. However, note that even compared to the VGG models without batch normalization, AlexNet is much less invariant and much more robust, together suggesting that shift invariance contributes to a lack of robustness.

In addition to these results on realistic CNNs, we also test Visual Transformers (ViTs), which do not have built in shift invariance. We compare 4 models that are pretrained on ImageNet21K and fine-tuned on ImageNet1K, and show the accuracy under attack in Figure 4a. The two ViT models are transformers, while the BiT models are ResNets with Group Normalization and standardized convolutions (Dosovitskiy et al., 2021). We can see that ViTs are much more adversarially robust than their CNN counterparts. This is contemporaneously found by Shao et al. (2021), also. We conjecture that this can be connected to the fact that ViTs do not have built in shift invariance. To test this, we calculate the consistency of these 4 models. The consistency of the ViT-B/16 and ViT-B/32 models are 83.51 and 78.06 respectively, while the consistency of BiT-ResNet50x1 and BiT-ResNet101x1 are 87.23 and 88.31 respectively. This indicates that visual transformers learn some measure of shift invariance, but are less shift invariant than BiT-ResNets. These less shift invariant ViT transformers therefore achieve better robustness than other architectures.

To further evaluate real-world architectures, we perform additional experments on CIFAR-10. We report test accuracy and robustness of these architectures under different attacks in Figure 5a. Unlike Imagenet where fully-connected networks don't perform well, it has been shown that FCNs can achieve fairly high accuracy on the CIFAR-10 dataset (Neyshabur, 2020). To compare the robustness

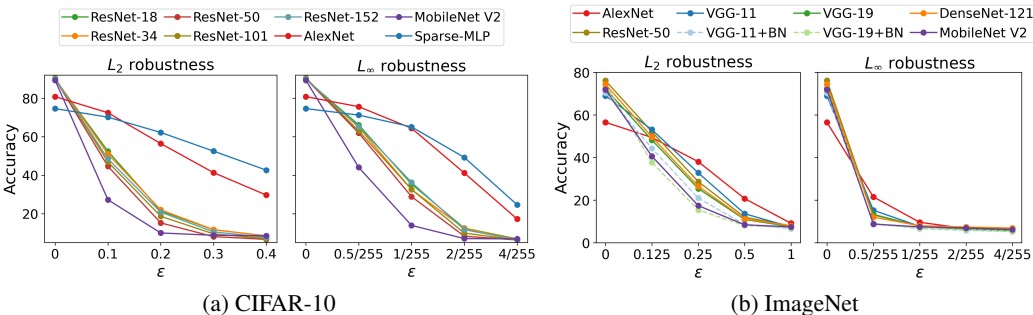

Figure 5: Robustness of models on CIFAR-10 and ImageNet datasets. "BN" indicates that the model uses batch normalization.

of FCNs we also train a 3-layer MLP network, named Sparse-MLP using the $\beta$-lasso method following the same experimental setup proposed by Neyshabur (2020).. For AlexNet and five variants of ResNet, we train the models for 200 epochs using SGD with a cosine annealing schedule (Loshchilov & Hutter, 2016). We use a momentum equal to 0.9 for SGD and weight decay of $5e - 4$. We then evaluate the consistency of the models to shifts and find that all the ResNet models have consistency larger than 75 while the consistency of AlexNet and Sparse-MLP is much smaller at 34.79 and 29.18 respectively. The clean accuracies of AlexNet and Sparse-MLP are lower than the ResNets. However, their robust accuracy decreases at a much slower rate as the attack strength increases, **showing that the less shift invariant classifiers, AlexNet and Sparse-MLP achieve better robustness**.

## 5.2 Dimension and margin

Training a network with a shift invariant architecture is similar to training a FC network using data augmentation with all shifted versions of the training set. This raises the question: why would data augmentation with shifted training examples affect adversarial robustness? Note that it has been argued that in a different context, more training data should lead to greater robustness (Schmidt et al., 2018). We examine this question with some inital experiments.

In prior work it has been suggested that larger input dimension of the data (Goodfellow et al., 2015; Gilmer et al., 2018) or larger intrinsic dimension of the data (Khoury & Hadfield-Menell, 2019) can reduce robustness. One way that shift invariance might reduce robustness is by increasing the intrinsic dimension of the training data.

To examine this question we create two synthetic datasets, one in which shift invariance increases the implicit dimension of the data, and one in which it does not. In both datasets, each class contains $n$, orthogonal vectors, with varying $n$ and dimension $d = 2000$. In the first set, the vectors are randomly chosen from a uniform distribution, subject to the constraint that they be orthogonal. The set of vectors in a single class have dimension $n$, but when we consider all shifted versions of even a single vector, this has has dimension $d$. In the second dataset, training examples are sampled by frequencies $\sin 2\pi kx$ and $\cos 2\pi kx$, where $k \in [1, n]$. The first class consists of frequencies in which $k$ is odd, and the second class consists of frequencies with even $k$. Because all shifts of $\sin 2\pi kx$ and $\cos 2\pi kx$ are linear combinations of these two vectors, the vectors in a class have dimension $n$, and all shifts of the vectors have dimension $n$.

If increasing the intrinsic dimension of the data reduces robustness, we would expect that for the FC network, increasing $n$ would reduce robustness for both datasets. However, we would expect that for the shift invariant network, we would see much lower robustness for dataset 1 than for dataset 2, especially for small $n$. We do observe this, as shown in Table 3.

In our theoretical results, and the example in Figure 1, we see that considering all shifts of the data may not only increase its dimension but also reduce the linear margin. To tease these effects apart, we create a third data set in which as its size increases, its dimension increases but its margin does not.

In this third data set, all samples in one class have a common component in a random direction, and a second component in a random direction that is orthogonal to all other vectors. That is, let $\mathbf{c}_1, \mathbf{c}_2, \mathbf{r}_i^1, \mathbf{r}_i^2, 1 \le i \le n$ be a set of randomly chosen, orthogonal unit vectors. Then the $i$'th vector in class $j$ is given by $\mathbf{x}_i^j = p\mathbf{c}_j + \mathbf{r}_i^j$, where $p$ is a constant that we vary. The margin between the classes will be at least $\sqrt{2}p$. Since we are not controlling the margin for all shifts of this data, we

| | Orth. Vectors | | Orth. Frequencies | |
| --- | --- | --- | --- | --- |
| n | FC | CNN | FC | CNN |
| 50 | 0.113 | 0.033 | 0.129 | 0.272 |
| 100 | 0.083 | 0.029 | 0.126 | 0.251 |
| 200 | 0.060 | 0.024 | 0.103 | 0.326 |

Table 3: Average $l_2$ distance for adversarial examples over all samples for dataset 1 and 2. For orthogonal vectors, FC networks are more robust than CNNs. For orthogonal frequencies, CNNs are more robust than FC networks.

| | $p$ | | | | | |
| --- | --- | --- | --- | --- | --- | --- |
| $n$ | 0 | 0.02 | 0.05 | 0.1 | 0.2 | 0.3 |
| 50 | 0.106 | 0.108 | 0.096 | 0.114 | 0.152 | 0.181 |
| 100 | 0.080 | 0.081 | 0.080 | 0.094 | 0.121 | 0.181 |
| 200 | 0.060 | 0.061 | 0.061 | 0.079 | 0.112 | 0.170 |

Table 4: Average $l_2$ distance for all samples on FC network for dataset 3. The average $l_2$ distance increases with and increase in $p$.

classify this data only using an FC network. As shown in Table 4, we observe that as $p$ increases, robustness increases. Also for large $p$, robustness is similar across different $n$ values suggesting that the linear margin predicts robustness better than the dimension of the data.

The experiments in this subsection are meant to highlight some interesting questions about the connection between shift invariance and robustness. Does shift-invariance reduce robustness by increasing the implicit dimension of the data? And if so, is this because it leads to data that implicitly has a smaller margin? Our tentative answer to these questions is yes, based on experiments with simple datasets. But these are questions that surely deserve greater attention.

**Potential Negative Societal Impacts**

Adversarial attacks on deep learning pose significant societal threats. They threaten to disrupt deployed machine learning systems, some of which may play a critical role in health and safety, including in health care systems or self-driving cars. Our work aims to provide a better fundamental understanding of these attacks. This might potentially lead to either better defenses against attacks or to more effective attacks. However, we feel that on balance, a better understanding of security threats is helpful in building more secure systems.

# 6 Conclusion

We have shown theoretically and experimentally that shift invariance can reduce adversarial robustness in neural networks. We prove that for linear classifiers, shift invariance reduces the margin between classes to just the difference in their DC components. Using this, we construct a simple, intuitive example in which shift invariance dramatically reduces robustness. Our experiments provide evidence that this reduction in robustness shows up in real networks as well. Finally, our experiments on synthetic data provide a step towards understanding the relationship between robustness, shift invariance, dimensionality and the linear margin of the data, in neural networks.

**Acknowledgments and Disclosure of Funding**

The authors thank the U.S.- Israel Binational Science Foundation, grant number 2018680, the National Science Foundation, grant no. IIS-1910132, the Quantifying Ensemble Diversity for Robust Machine Learning (QED for RML) program from DARPA and the Guaranteeing AI Robustness Against Deception (GARD) program from DARPA for their support of this project.

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
