# OpenReview forum: "Shift Invariance Can Reduce Adversarial Robustness"
_NeurIPS.cc/2021/Conference — NeurIPS 2021 Poster_

### Official Review · Reviewer_7xCn · 2021-07-14

**Rating:** 3
**Confidence:** 3

**Summary:**

This paper begins with the finding that shows that shift-invariant linear classifiers can have a significantly smaller margin than non-shift-invariant linear classifiers. They then perform a similar analysis on neural tangent kernels (NTK) and find that the linear classifier trained on the convolutional NTK also has a smaller margin. Empirically, they found that some networks, like AlexNet, are less shift-invariant and they seem to be more robust.- The observation that shift Invariance can reduce adversarial robustness is an interesting and important topic.
- The experimental results are worth a deeper analysis of why less shift-invariant networks appear more robust.
- Code is provided in the supplementary.


**Limitations And Societal Impact:**

There are several limitations of this work not mentioned by the authors:
- There gap between theory and practice is too large and the theorem cannot represent most of the particle cases.
- Many confounding factors mentioned in my review are not discussed in the paper.

I am not aware of any potential negative social impact of this paper.

**Main Review:**

Pros:
- The observation that shift Invariance can reduce adversarial robustness is an interesting and important topic.
- The experimental results are worth a deeper analysis of why less shift-invariant networks appear more robust.
- Code is provided in the supplementary.

Cons:

(1) The theoretical analysis is straightforward. In both the analysis in sections 3 and 4, a constraint on being shift-invariance is added to the classifier. When a constraint is added, the margin can only be smaller or equal to the original one.

(2) The gap between theory and practice is too large.
 - The theoretical analysis can only show that the margin decreases when the shift-invariant constraint is added. However, in practice, when we say adversarial example we only talk about ``small'' perturbation. As long as there is a healthy amount margin, it should be fine. So the real question is, does enforcing shift-invariance make the margin too small so that adversarial examples are inevitable.
 - If there is an example of an image being shifted into the adversarial example of an image from another class, this paper would be much more interesting.

(3) The authors mentioned that this work is quite relevant to (Kamath et al. (2020)). So how is this work differently from the work by Kamath et al.? Are there any new findings?

(4) The empirical evidence shown here is not enough to convince me that ``in general'', more shift-invariant network will be less robust:
 - If two networks are trained with the same architecture and optimizer, one network is trained with shifting data augmentation and the other one is trained without data augmentation is the result still consistent with your findings. Suggestion: more experiments like section 5.1.2 should be done.
 - It is mentioned that some pre-trained models are retrieved from Pytorch model zoo on ImageNet. In the comparison, are the models all trained with the same data augmentation method? I think this should be discussed as this can be a confounding factor that may change the interpretation of the result.
 - The networks in each figure are not consistent. In Figure 4, why Sparse-MLP appeared in Cifar10 but not in ImageNet? why DenseNet, MobileNet does not appear in Cifar10?
 - Does the findings still hold when network trained with adversarial training? If adversarial training is used, does the level of shift-invariance decrease?
 - It appears models with lower natural accuracy also more robust? Could it be because that AlexNet is less overfitting to the data so that it is more robust?

(5) "We have shown theoretically that shift invariance can reduce adversarial robustness." I don't agree with this statement in line 197 as sections 3 and 4 only show that shift invariance can reduce the margin. It does not say the margin becomes too small. If you have a super large perturbation and the perturbed image looks completely different from the original one. Even if this perturbed image is predicted as a different class, it wouldn't necessarily be called an adversarial example.

minor:
- The figures can be organized in a way that they are on the same page with the text that is mentioning the figure.
- C&W attack should be used for L2 attacks https://arxiv.org/abs/1608.04644.

**Time Spent Reviewing:**

5

---

> ### Author Response · Authors · 2021-08-10
> **Response to Reviewer 7xCn**
>
>
> We thank the reviewer for their comments. In the following text, we address the reviewer's concerns -
>
> *"The theoretical analysis is straightforward. In both the analysis in sections 3 and 4, a constraint on being shift-invariance is added to the classifier. When a constraint is added, the margin can only be smaller or equal to the original one."*
>
> *"The theoretical analysis can only show that the margin decreases when the shift-invariant constraint is added. However, in practice, when we say adversarial example we only talk about ``small'' perturbation. As long as there is a healthy amount margin, it should be fine. So the real question is, does enforcing shift-invariance make the margin too small so that adversarial examples are inevitable."*
>
> *"If there is an example of an image being shifted into the adversarial example of an image from another class, this paper would be much more interesting."*
>
> In several places, this reviewer suggests that our theoretical results only show that shift invariance reduces the margin of the classifier, but that our results do not quantify the reduction in margin.  Points 1 and 2 of the reviewer imply this, with the reviewer stating: “So the real question is, does enforcing shift-invariance make the margin too small so that adversarial examples are inevitable” In fact, for the adversarial example we consider, we prove that for an image of size $d$, the margin shrinks as $1/\sqrt(d)$.  Figure 1 shows that the margin becomes quite small for images of width 10 or larger.  This Figure shows two adversarial examples for the shift-invariant network which appear nearly identical to the original images, but have different labels.  Here Figure 1a shows the original images, and Figure 1b, column 1, labeled “CNN” shows the adversarial examples, which have different labels.  We feel that this addresses the comment: “If there is an example of an image being shifted into the adversarial example of an image from another class, this paper would be much more interesting.”  We apologize if it was not clear that Figure 1 shows exactly such an example.
>
> *"The authors mentioned that this work is quite relevant to (Kamath et al. (2020)). So how is this work different from the work by Kamath et al.? Are there any new findings?"*
>
> (Kamath et al. (2020)) examines the interplay between rotation invariance and robustness. They empirically show that by inducing more rotational invariance using data augmentation in CNN's and Group-equivariant CNN's the adversarial robustness of these models decreases. They also show that for adversarially trained models, the rotation invariance for the models decreases as $\epsilon$ used for the $l_{\infty}$ adversarial training increases.
>
> *"If two networks are trained with the same architecture and optimizer, one network is trained with shifting data augmentation and the other one is trained without data augmentation is the result still consistent with your findings. Suggestion: more experiments like section 5.1.2 should be done."*
>
> We note that our work addresses shift-invariance that is due to the choice of network architecture, not data augmentation.  If we include all shifts of the training data, the training set becomes huge.  Using a sample of these shifts for data augmentation, we have found it difficult to achieve shift-invariance.  However, we stress that our paper does not address the issue of data augmentation.
>
> *"It is mentioned that some pre-trained models are retrieved from Pytorch model zoo on ImageNet. In the comparison, are the models all trained with the same data augmentation method? I think this should be discussed as this can be a confounding factor that may change the interpretation of the result."*
>
> Again, we would like to stress that none of our experiments use data augmentation.  Some of the models from the Pytorch model zoo are naturally more shift-invariant than others (eg., Alexnet vs. ResNet), and we measure this using consistency.  But all our experiments use the pre-trained models in the model zoo.
>
> *"The networks in each figure are not consistent. In Figure 4, why Sparse-MLP appeared in Cifar10 but not in ImageNet? why DenseNet, MobileNet does not appear in Cifar10?"*
>
> We do not run experiments with Sparse-MLP on ImageNet because it is not able to work well on this data set.  To our knowledge, no MLP-based model is able to handle ImageNet.  To increase the consistency of our experiments, as suggested by the reviewer, we have run experiments applying DenseNet and MobileNet to CIFAR-10.  These models show similar behavior to ResNet and VGG as shown below.
>
> |              | clean  | L2-0.1 | L2-0.2 | L2-0.3 | L2-0.4 | L∞-1/510 | L∞-1/255 | L∞-2/255 | L∞-3/255 |
> |--------------|--------|--------|--------|--------|--------|----------|----------|----------|----------|
> | AlexNet      | 80.81% | 72.40% | 56.25% | 41.12% | 29.85% | 77.24%   | 69.08%   | 49.67%   | 23.34%   |
> | ResNet-50    | 90.12% | 44.39% | 15.11% | 8.06%  | 6.60%  | 69.89%   | 40.19%   | 12.08%   | 6.36%    |
> | DenseNet-121 | 92.14% | 54.13% | 33.52% | 22.64% | 16.44% | 65.43%   | 44.68%   | 24.22%   | 10.56%   |
> | MobileNet    | 89.32% | 27.23% | 9.96%  | 8.77%  | 8.52%  | 44.05%   | 13.90%   | 7.04%    | 6.88%    |
>
> *"Does the findings still hold when network trained with adversarial training? If adversarial training is used, does the level of shift-invariance decrease?"*
>
> We appreciate this suggestion.  We have measured the consistency of PreActResnet18 before and after adversarial training using PGD. We used PreActResnet18 since it trains faster than other SOTA networks and achieves high robustness after adversarial training [1]. We find that adversarial training reduces the consistency from 68.93 to 46.63.  So adversarial training does indeed reduce shift-invariance.
>
> *"It appears models with lower natural accuracy also more robust? Could it be because that AlexNet is less overfitting to the data so that it is more robust?"*
>
> This is an interesting suggestion.  However, we note that architectures using transformers achieve high accuracy on ImageNet with less shift-invariance.  It is difficult to draw a firm conclusion from these examples, but they bear investigation.
>
> *""We have shown theoretically that shift invariance can reduce adversarial robustness." I don't agree with this statement in line 197 as sections 3 and 4 only show that shift invariance can reduce the margin. It does not say the margin becomes too small. If you have a super large perturbation and the perturbed image looks completely different from the original one. Even if this perturbed image is predicted as a different class, it wouldn't necessarily be called an adversarial example."*
>
> Please see our first response above.  We do in fact show that shift invariance can reduce the margin to zero, in the limit of large images.
>
> [1] "Fixing Data Augmentation to Improve Adversarial Robustness", Sylvestre-Alvise Rebuffi et al. arXiv preprint arXiv:2103.01946

---

> > ### Author Response · Authors · 2021-09-01
> > **Lack of response**
> >
> > We feel that this review was based on a significant misunderstanding of the results presented in our paper.  Most importantly, the reviewer seems to think that we did not discuss situations in which small perturbations lead to adversarial examples.  We have tried in our initial response to clarify any confusing points, in particular explaining how the paper deals directly with this issue, and shows examples in which imperceptible perturbations can change the class of an image.  We are disappointed that the reviewer has not engaged in a dialog with us to ensure that the review is based on an accurate understanding of our results.

---

### Official Review · Reviewer_SjMk · 2021-07-15

**Rating:** 6
**Confidence:** 4

**Summary:**

The adversarial attack is one of the critical problems in current machine learning research that causes end-users to mistrust models deployed for high-stakes decision-making applications. The paper identifies the "shift-invariance" property of neural networks to be a possible reason behind their high sensitivity to adversarial attacks. The paper shows that the margin between the classes for a shift-invariant linear classifier only depends on the DC component of the signals. Further, the authors show that in simple cases,
fully connected and shift-invariant neural networks learn linear decision boundaries. Finally, experimental evaluations on real datasets and architectures show that shift invariance reduces adversarial robustness.

**Limitations And Societal Impact:**

There are a few open questions still remaining which have been addressed by the authors themselves. The authors present a very interesting idea which is very important to develop and deploy trustworthy machine learning models but it needs further explanation.

**Main Review:**

Strengths:
1. The paper is very well-motivated and addresses an important problem in adversarial machine learning, i.e., a better understanding of adversarial attacks.
2. The work provides a better understanding of the relationship between adversarial robustness, shift-invariance, dimensionality, and the linear margin of the data in neural networks.
3. The hypothesis for shift-invariance is empirically tested on large datasets and diverse neural network architectures such as FC, CNNs, and transformers.


Open concerns:
1. It would be helpful if the authors can comment on why for easier datasets like Cifar-10, we observe that AlexNet is an outlier both in terms of shift-invariance and robustness, but for harder datasets like ImageNet, the gap reduces significantly (Fig. 4).
2. The connection between shift-invariance and robustness is interesting but unclear. The experimental results show some correlation between the two, but the hypothesis that shift invariance reduces robustness by increasing the implicit dimension of the data is not well explained.
3. From Fig. 2, the authors conclude that "fully-connected networks (FCNs) are more robust than Shift Invariant CNNs". It would be beneficial if the authors can provide the shift-invariance score of the respective networks to support their conclusion. In addition, unlike FCNs, convolutional neural networks (CNNs) have a spatial bias -- a type of inductive bias that assumes a certain spatial structure present in the input data. Can this contribute to their poor adversarial robustness performance?
4. In Fig. 1, we observe that the margin of a linear, shift-invariant classifier will shrink in proportion to the inverse of the total number of image pixels. It is interesting because most pixels in the images from Fig. 1 belong to the same color (gray). It would be great if the authors can comment on this answer.

**Time Spent Reviewing:**

10

---

> ### Author Response · Authors · 2021-08-10
> **Response to Reviewer SjMk**
>
> We thank the reviewer for their comments and the time spent reviewing our work. In the following text, we address the reviewer's concerns -
>
> *"It would be helpful if the authors can comment on why for easier datasets like Cifar-10, we observe that AlexNet is an outlier both in terms of shift-invariance and robustness, but for harder datasets like ImageNet, the gap reduces significantly (Fig. 4)."*
>
> The reviewer raises an interesting question.  We speculate that because ImageNet (1.3M) has much more training data than CIFAR-10 (50K), perhaps with some implicit shifts, models without a strong implicit bias on shift-invariance like AlexNet may learn some invariance from the data.  But this question merits further investigation.
>
> *"The connection between shift-invariance and robustness is interesting but unclear. The experimental results show some correlation between the two, but the hypothesis that shift invariance reduces robustness by increasing the implicit dimension of the data is not well explained."*
>
> We regret that our explanation of this point, and the experiments in Table 3, were not clearer.  We tested the hypothesis that shift invariance reduces robustness using synthetic data.  We created two datasets, call them $D_1$ and $D_2$.  Let $SH(D_1)$ denote the set of all possible shifts of all images in $D_1$.  Define $SH(D_2)$ similarly.  We created our data so that the dimension of $SH(D_1)$ was much larger than the dimension of $D_1$, but the dimension of $SH(D_2)$ was the same as the dimension of $D_2$.  Then we found that using a shift-invariant architecture reduced robustness when $D_1$ was used as the training set, but shift-invariance did not affect robustness for $D_2$.  This suggests that shift invariance only reduces robustness when it increases the implicit dimension of the training data.  We acknowledge, though, that while suggestive, this experiment is only a starting point for examining this question.
>
> *"From Fig. 2, the authors conclude that "fully-connected networks (FCNs) are more robust than Shift Invariant CNNs". It would be beneficial if the authors can provide the shift-invariance score of the respective networks to support their conclusion. In addition, unlike FCNs, convolutional neural networks (CNNs) have a spatial bias -- a type of inductive bias that assumes a certain spatial structure present in the input data. Can this contribute to their poor adversarial robustness performance?"*
>
> We agree that the relative lack of robustness of CNNs may be due to its spatial bias, as we feel that this bias is intimately related to the approximate shift-invariance of CNNs.  We have computed the consistency of the fully connected networks we train, shown below.  We can see that the degree of shift-invariance of these networks is very low.
>
> | MNIST FCN | Consistency | F-MNIST FCN | Consistency |
> |-------------------------|-------------|---------------------------|-------------|
> | FC-256                  | 16.00       | FC-2-256                  | 19.46       |
> | FC-512                  | 20.69       | FC-3-256                  | 20.68       |
> | FC-1024                 | 20.69       | FC-2-512                  | 18.19       |
> | FC-2048                 | 19.20       | FC-3-512                  | 18.38       |
>
> *"In Fig. 1, we observe that the margin of a linear, shift-invariant classifier will shrink in proportion to the inverse of the total number of image pixels. It is interesting because most pixels in the images from Fig. 1 belong to the same color (gray). It would be great if the authors can comment on this answer."*
>
> The behavior shown in Figure 1 will be observed in any two images that are anti-podal (one is the negative of the other) and in which the DC component of the images goes to zero as the image size increases.  For example, if one image was a Gaussian of fixed variance, and the second image was the negative of this image, our theory would also apply, and shift-invariance would lead to adversarial examples arbitrarily close to the initial images.  In this case, the pixels would not be purely gray, although they would be close to gray as the image becomes large.

---

> > ### Comment · Reviewer_SjMk · 2021-08-25
> > **Discussion response**
> >
> > Dear authors,
> >
> > Thank you for your detailed response. Here are some of my clarification thoughts:
> >
> > 1) As agreed by the authors, the synthetic data experiment is indicative but not significant to suggest that shift invariance only reduces robustness when it increases the implicit dimension of the training data.
> >
> > 2) I observe that the degree of shift-invariance of these networks is low, but the consistency score is also similar for different architectures. I would have expected some trend between the model complexity and consistency score.

---

> > > ### Author Response · Authors · 2021-08-27
> > > **Response to comment**
> > >
> > > Thanks for your comments:
> > >
> > > 1) Yes, we agree.  We have shown one experiment in which manipulating the effect of shift invariance on the implicit data dimension leads to significant effects on adversarial robustness.  We believe that this results suggests the promise of more extensive future experiments.
> > >
> > > 2) Indeed, it has been found (https://github.com/adobe/antialiased-cnns.) that for CNNs, model complexity correlates with shift invariance.  We do not find this to be the case for fully connected networks.  Presumably this is because these are highly non-shift invariant for both simple and more complex models.

---

### Official Review · Reviewer_UPzX · 2021-07-16

**Rating:** 6
**Confidence:** 3

**Summary:**

This work provides theoretical and empirical justifications on why shift-invariant neural networks are less adversarial robust than fully-connected neural networks. Using Neural Tangent Kernels and Convolutional Neural Tangent Kernels, the authors prove that for simple neural networks, shift-invariant neural networks produce separators with decreasing margins, making them not as robust as fully-connected neural networks. Empirically, the authors provide experiments on simple neural networks with MNIST and more complicated networks such as ResNet on bigger datasets such as CIFAR10.

**Ethical Concerns:**

There are no ethical concerns with this work.

**Limitations And Societal Impact:**

Despite the pros above, I am not entirely convinced it is a fair problem to compare adversarial robustness of shift-invariant networks vs fully-connected networks. For both theory and experiments, the tasks designed for the two networks are different. For instance, facial recognition that requires translation invariance is a different task than facial recognition with aligned images. What might be a better comparison in this paper is if the authors only focus on shift-invariant networks, rather than comparing fully-connected networks to shift-invariant networks.

In Section 5.2, the authors compared “training a shift-invariant network is similar to training a fully-connected network using all shifted versions of the training set.” This is true generally, at the very least, experiment results should show that the network trained is actually shift-invariant. Or else, the experiment does not seem convincing that it actually applies to shift-invariant networks.

In addition, since the network now is shift-invariant, to test adversarial robustness, it seems to deserve a more thorough study on doing experiments on generating adversarial images with not only original images but also shifted images.

Arguably, requiring a network to be shift-invariant also imposes a harder challenge to the learning problem than training a network without shift-invariance. I would also expect that imposing more shift-invariance on a network will naturally lead to weaker performances and worse adversarial robustness. Although this work provides some good experiments to verify this, I am not sure if the work’s contribution is big enough to be accepted for this conference.

If the authors can provide a better argument for why the comparison is fair, or point out any important parts I misunderstood, I am more than happy to change my score.

The authors have adequately address the work's societal impact.

**Main Review:**

There is originality in this work; however, there are a number of works that address the relationship between invariance and robustness in the past. Related works also seem to be adequately cited. The submission is also well-organized, with good teaser figures and example in the front, followed by a number of theorems and experiments to justify the claims made.

There are a wide-range of experiments, testing different architectures and datasets. This shows that authors are willing to testify their claims through their experiments carefully. Having theorems to justify those claims is also a plus usually.

**Time Spent Reviewing:**

7

---

> ### Author Response · Authors · 2021-08-10
> **Response to Reviewer UPzX**
>
> We thank the reviewer for their comments and feedback on our work. In the following text, we address the reviewer's concerns individually -
>
> *"Despite the pros above, I am not entirely convinced it is a fair problem to compare adversarial robustness of shift-invariant networks vs fully-connected networks....For instance, facial recognition that requires translation invariance is a different task than facial recognition with aligned images. What might be a better comparison in this paper is if the authors only focus on shift-invariant networks, rather than comparing fully-connected networks to shift-invariant networks."*
>
> We agree with the reviewer that the value of shift-invariance may be quite task-dependent.  Certainly, CNNs have proven very effective in many applications.  We believe that it is generally held that part of the reason for the effectiveness of CNNs is that the inductive bias introduced by nearly shift-invariant convolutions is beneficial in these applications.  Our goal is to study the effect of the shift-invariance in these architectures on adversarial robustness.  We feel that the best way to understand these effects is by contrasting them with architectures that are not shift-invariant, such as fully connected networks.  It is not our intention to argue that one type of architecture is better than another, but rather to understand the effect of architectural choices on adversarial robustness.
>
> *"In Section 5.2, the authors compared “training a shift-invariant network is similar to training a fully-connected network using all shifted versions of the training set.” This is true generally, at the very least, experiment results should show that the network trained is actually shift-invariant. Or else, the experiment does not seem convincing that it actually applies to shift-invariant networks."*
>
> We wish to emphasize that the quoted remark is designed to provide intuitions about the effect of shift-invariant architectures.  None of our experiments involved training with shifted versions of the training set.  In the experiments shown in Figure 2, the CNN's used were designed to be provably, perfectly shift-invariant.  In the experiments shown in Tables 1 and 2, we methodically varied the amount of shift-invariance induced by the architecture and showed that reducing shift-invariance increased robustness.  In the experiments shown in Figure 3, we examined the behavior of real-world networks and measured the degree of their shift-invariance, showing that greater shift-invariance correlated with less robustness.  In all cases, we either built-in perfect shift-invariance or measured the degree of shift-invariance.
>
> *"In addition, since the network now is shift-invariant, to test adversarial robustness, it seems to deserve a more thorough study on doing experiments on generating adversarial images with not only original images but also shifted images."*
>
> In cases in which a network is perfectly shift-invariant, the network response will be the same for all shifted copies of a signal. So shifted versions of adversarial examples will also be adversarial.
>
> *"Arguably, requiring a network to be shift-invariant also imposes a harder challenge to the learning problem than training a network without shift-invariance. I would also expect that imposing more shift-invariance on a network will naturally lead to weaker performances and worse adversarial robustness. Although this work provides some good experiments to verify this, I am not sure if the work’s contribution is big enough to be accepted for this conference."*
>
> We would like to reiterate that our paper discusses shift-invariance that results from network architecture choices. In real networks, some degree of shift-invariance results from the use of CNNs, particularly those without FC layers, such as ResNet. It is our impression that network architectures have evolved to have more shift-invariance and better performance (eg., ResNet vs. Alexnet), at least until the recent popularity of transformer architectures. So greater shift-invariance has generally been associated with improved performance. We are not aware of prior publications that argue that CNNs would be expected to be less robust than less shift-invariant architectures. We are not sure why this would be naturally expected.

---

> > ### Comment · Reviewer_UPzX · 2021-09-01
> > **Thank you for clarification**
> >
> > Thank you for your detailed response. I realize the main confusion comes from this difference between working with CNN's which aims to be shift-invariant versus the CNN's you provided in the experiments which are provably shift-invariant. Then the line of thought "shifted versions of adversarial examples will also be adversarial" follow naturally. I think it would be an interesting direction to combine the type of analysis done here in this work with other provably invariant networks such as scattering networks.

---

### Official Review · Reviewer_yPtR · 2021-07-16

**Rating:** 6
**Confidence:** 3

**Summary:**

They show that invariance to circular shifts increase adversarial vulnerability. To motivate this finding, they characterize the margin between classes for a shift-invariant linear classifier. Based on the NTK settings, they show that FC and shift-invariant networks produce linear decision boundaries, which partially explains their adversarial vulnerability. They further empirically verify that shift-invariance reduces adversarial robustness on real datasets and realistic architectures.

**Limitations And Societal Impact:**

Yes.

**Main Review:**

Strengths
- Their hypothesis about shift invariance is first verified in a simple setting. They also provide theoretical justification and further demonstrate the evidence in real-world architectures and datasets.

Weaknesses
- I found this is a nice paper. I don't particularly see weaknesses.

Minor:
L292: Visual Transformer -> Vision Transformer

Additional comments:
- It would be interesting to consider more common image corruptions like gaussian noise, blur, etc.

**Time Spent Reviewing:**

3

---

> ### Author Response · Authors · 2021-08-10
> **Response to Reviewer yPtR**
>
> We thank the reviewer for their positive comments.
>
> **It would be interesting to consider more common image corruptions like Gaussian noise, blur, etc.** - We agree that it would be interesting to study the effects of invariance to other operations, such as blur, which are usually achieved through data augmentation.  Invariance to blur would effectively mean that the high-frequency components of the image are ignored, projecting all training and test data to a somewhat lower-dimensional space.  It would be interesting to see if this reduction in the intrinsic dimension of the data leads to an increase in robustness.

---

### Author Response · Authors · 2021-08-10
**Response to All Reviewers**

We thank the reviewers for their thoughtful comments.  We wish to emphasize one point that might not have come across clearly in the paper.  We are examining the effect of network architecture on adversarial robustness.  For example, a simple CNN with cyclic padding followed by global average pooling is completely shift-invariant; regardless of how it is trained, it will respond identically to inputs that are shifted versions of each other.  A fully connected network has no built-in shift-invariance.  Most real-world CNNs are not fully shift-invariant but are nearly so in practice.  We wish to emphasize that our paper does not directly study the effects of data augmentation.  Data augmentation aimed at promoting shift-invariance should have similar effects to shift-invariance that is built into the network architecture.  But none of our experiments or theory rely on data augmentation.  This allows us to study the effects of shift-invariance, without confounding factors from augmentation, such as the fact that augmentation methods also use more data [1], that augmentation itself may be a confounding factor [2].

[1] http://proceedings.mlr.press/v119/chen20q.html

[2] https://arxiv.org/abs/2103.01946

---

### Decision · Program_Chairs · 2021-09-27

**Decision:**

Accept (Poster)

**Comment:**

This paper provides theoretical and empirical evidences that the robustness of a (linear) classifier reduces when one tries to ensure shift invariance for the input. The results are not as surprising since classifying all shifted instances is indeed conceptually more difficult, when the feature dimension is fixed. The analysis is done for two-layer networks. One of the reviewers raised a legitimate concern whether the notion of robustness studied here is directly associated with the adversarial robustness. Majority of the reviewers nevertheless think the paper is above the standard for acceptance.